# Don't Just Prune by Magnitude! Your Mask Topology is Another Secret Weapon

**Duc Hoang**
University of Texas at Austin
`hoangduc@utexas.edu`

**Souvik Kundu**
Intel Labs
`souvikk.kundu@intel.com`

**Shiwei Liu**
Eindhoven University of Technology
University of Texas at Austin
`s.liu3@tue.nl`

**Zhangyang Wang**
Univerity of Texas at Austin
`atlaswang@utexas.edu`

## Abstract

Recent years have witnessed significant progress in understanding the relationship between the connectivity of a deep network's architecture as a graph, and the network's performance. A few prior arts connected deep architectures to expander graphs or Ramanujan graphs, and particularly,[7] demonstrated the use of such graph connectivity measures with ranking and relative performance of various obtained sparse sub-networks (i.e. models with prune masks) without the need for training. However, no prior work explicitly explores the role of parameters in the graph's connectivity, making the graph-based understanding of prune masks and the magnitude/gradient-based pruning practice isolated from one another. This paper strives to fill in this gap, by analyzing the *Weighted Spectral Gap* of Ramanujan structures in sparse neural networks and investigates its correlation with final performance. We specifically examine the evolution of sparse structures under a popular dynamic sparse-to-sparse network training scheme, and intriguingly find that the generated random topologies inherently maximize Ramanujan graphs. We also identify a strong correlation between masks, performance, and the weighted spectral gap. Leveraging this observation, we propose to construct a new "full-spectrum coordinate" aiming to comprehensively characterize a sparse neural network's promise. Concretely, it consists of the classical Ramanujan's gap (structure), our proposed weighted spectral gap (parameters), and the constituent nested regular graphs within. In this new coordinate system, a sparse subnetwork's $\ell_2$-distance from its original initialization is found to have nearly linear correlated with its performance. Eventually, we apply this unified perspective to develop a new actionable pruning method, by sampling sparse masks to maximize the $\ell_2$-coordinate distance. Our method can be augmented with the "pruning at initialization" (PaI) method, and significantly outperforms existing PaI methods. With only a few iterations of training (e.g 500 iterations), we can get LTH-comparable performance as that yielded via "pruning after training", significantly saving pre-training costs. Codes can be found at: `https://github.com/VITA-Group/FullSpectrum-PAI`.

## 1 Introduction

Pruning [21] reduces the size of deep neural networks (**DNNs**) by generating sparse models suitable for compute and memory-limited applications while still preserving comparable accuracy as their dense counterparts. Existing research in this area can broadly be divided into three main components. Firstly, pruning after training (**PaT**) that involves creating sparse DNNs by leveraging information

37th Conference on Neural Information Processing Systems (NeurIPS 2023).

from a trained dense model. The Lottery Ticket Hypothesis (**LTH**) [4], a notable work in PaT, suggests a sparse subnetwork within a dense trained model can achieve comparable performance. Based on this hypothesis, they propose iterative magnitude pruning (IMP) to identify such "lottery" subnetwork at an expensive cost. Secondly, Dynamic Sparse Training (**DST**) [14, 11, 2] starts with training a sparse network from scratch instead of using a pre-trained dense model. Here, the sparsity mask keeps on updating throughout the training to reach maturity while maintaining the target sparsity during each epoch, often referred to as "sparse-to-sparse" training [12]. The mask update in DST primarily relies on pruning and regrowing operations. Finally, Pruning at Initialization (**PaI**) [9, 20, 18] keeps the sparsity mask frozen throughout the training Various proxies have been proposed [9, 20, 18, 2] to identify the sparsity mask before training, although the PaI-yielded performance so far falls clearly below LTH and other PaT methods [6] despite the former's appealing low overheads.

Despite the significant overlap between these methods, particularly in their use of various proxies like magnitude, they differ in their beliefs on the origin of the inherent knowledge represented by the proxy used to create the sparsity mask. PaI assumes that the knowledge emerges during the initialization phase, while PaT and DST hypothesize that the knowledge is acquired from a fully or partially trained model. Empirical studies ([16], [6]) have demonstrated the effectiveness of PaT over PaI under high sparsity conditions. However, such performance benefits often come at the cost of high training requirements due to the iterative nature of PaT. Interestingly, a recent work [10] highlighted that even randomly pruned subnetworks can achieve similar performance as the dense model, further raising questions about the applicability of different proxies. These often conflicting and counter-intuitive results intensify the debate of proxy knowledge generation through matured training vs. that identified at the beginning.

To unravel this mystery, we draw inspiration from a complementary perspective of graph theory to precisely understand the relationship between the connectivity of a neural network's architecture as a graph and the network's performance. Specifically, we focus on a subset of the expander graph family, namely the Ramanujan graphs. Ramanujan graphs have a maximal eigenbound, allowing the network to be highly sparse and highly connected. This aligns with the primary objective of pruning — to find a sparse yet highly connected network. Previous works [19, 15, 1, 17, 7] have provided empirical evidence supporting this intuition by utilizing the Ramanujan property to guide existing sparse generators. However, to our understanding, there exists several missing links:

1. *Is the formation of the Ramanujan characteristic in sparse structures a natural occurrence during training? If so, how can we observe and quantify it?*

2. *What is the role of weight magnitudes in a Ramanujan graph and can they have a relation with the graph topology? If so, can there be a unified representation to encompass both?*

Towards closing this gap, we first investigate the evolution of sparse structures under In-Time Over-Parameterization (ITOP) [13], a popular DST regime using magnitude pruning and random growth. Interestingly, we observe that DST's random growth is inherently a maximizer of Ramanujan graphs (though not a very efficient one). We then discover a negative correlation between the performance and the weighted spectral gap ($\lambda$). To further leverage this observation, we propose to construct a new "full-spectrum coordinate" aiming to comprehensively characterize a sparse neural network's promise, by combining Ramanujan's bound $\Delta r$ (structure), weighted spectral gap $\lambda$ (parameters), and the constituent nested regular graphs within (as in the case of $\Delta r_{imdb}$ [7]). Most strikingly, in the resultant coordinate system, we find that a sparse sub-network's $\ell_2$-moving distance from its original initialization has a nearly linear correlation with the network's performance. Inspired by this, we maximize this $\ell_2$-distance in the coordinate by greedily sampling sparse masks at initialization, yielding a new PaI approach. We empirically verify that our new PaI method, dubbed *Pruning at Initialization as Graph Sampling* (**PAGS**), can create significantly better "zero-shot" masks than the existing PaI methods. Our contributions are outlined as:

- We first uncover how a sparse topology is evolved in a representative dynamic sparse training scheme (ITOP) by analyzing the weighted spectral gap of Ramanujan structures. We discover that this mechanism is an inherent yet inefficient Ramanujan maximizer.

- We then establish a full-spectrum coordinate to jointly measure the structure and weight distance by combining the Ramanujan perspective with the weighted spectral gap, demonstrating a sparse subnetwork's $\ell_2$-moving distance from its initialization in the new coordinate as a strong (linear) performance indicator.

- We propose *Pruning at Initialization as Graph Sampling* (PAGS) by greedily maximizing the aforementioned $\ell_2$-moving distance. PAGS can be organically applied as a "zero-shot" PaI method and outperforms existing PaI methods with large margins. With only a few iterations of training (e.g. only 500 iterations), we can get LTH-comparable performance as that yielded via IMP in PaT, significantly saving pre-training costs.

## 2 Observing Weight and Topology Evolution via Dynamic Sparse Training

### 2.1 Notations and Definitions

We first introduce two important concepts that are used to explain key observations made later on in this section. We also summarized the various notations in Table 1 for easy reference.

**Ramanujan gap:** Belonging to a subset of the expander graphs, Ramanujan graphs are distinguished by their sparse yet highly interconnected nature. High connectivity in a graph implies smoother information flow, a trait coveted in sparse neural networks. Previous research [19, 15, 1, 17] interpreted DNNs as a series of bipartite compute graphs (check the appendix for more details). Here, each layer takes the form of a square adjacency matrix $A$. Pruning strategies, inspired by Ramanujan properties, aim to widen the gap between Ramanujan's upper-bound, $2 * \sqrt{d-1}$, where $d$ is the average number of edge per node, and the non-trivial eigenvalue of the compute adjacency matrix, $\hat{\mu}(A)$. This gap can be viewed in two ways:

- Canonical perspective: $\Delta r = 2 * \sqrt{d-1} - \hat{\mu}(A)$
- Iterative perspective: $\Delta r_{imdb} = \frac{1}{|K|} \sum_{i=1}^{|K|} (2\sqrt{d_i - 1} - \hat{\mu}(A_{K_i}))$

While the canonical perspective, $\Delta r$, indicates the ease of information propagation by measuring the network's degree of connectivity, a more recent take by [7] introduces the iterative perspective, $\Delta r_{imdb}$. This measures the average connectivity limit across all subgraphs $K$ within $A$, potentially offering a more comprehensive insight into the network's connectivity. Note that "imdb" stands for "iterative mean difference of bound", following the same naming scheme in [7].

**Weighted spectral gap:** Of our own devise, the weighted spectral gap, denoted as $\lambda$, quantifies the separation between the trivial ($\mu_0$) and non-trivial ($\hat{\mu}$) eigenvalues of the weighted adjacency matrix, $\boldsymbol{W}$. $\lambda$ provides insights analogous to the Cheeger constant, which evaluates if a graph possesses a "bottleneck". Mirroring the two previous perspectives for Ramanujan gap, the weighted spectral gap is similarly dual-formed:

- Canonical perspective: $\lambda = \mu_0(|\boldsymbol{W}|) - \hat{\mu}(|\boldsymbol{W}|)$,
- Iterative perspective: $\lambda_{imsg} = \frac{1}{|K|} \sum_{i=1}^{|K|} \left( \mu_0(|\boldsymbol{W}_{K_i}|) - \hat{\mu}(|\boldsymbol{W}_{K_i}|) \right)$

In this context, $W$ stands for the layer-specific weight elements. However, unlike $\Delta r$ and $\Delta r_{imdb}$, the interpretations of $\lambda$ and $\lambda_{imsg}$ are not universally settled in relation to the network's sparsity and their performance correlation. However, later on in this section, we reveal them to be reliable indicators for performance. Note, here "imsg" stands for "iterative mean spectral gap."

Table 1: Important graph notations, their equations, and descriptions.

| Notation | Equation | Description |
|---|---|---|
| $\Delta r$ | $2 * \sqrt{d-1} - \hat{\mu}(A)$ | Measure a graph's degree of connectivity using Ramanujan bound |
| $\Delta r_{imdb}$ | $\frac{1}{|K|} \sum_{i=1}^{|K|} (2\sqrt{d_i - 1} - \hat{\mu}(A_{K_i}))$ | Iterative mean of Ramanujan gap for set of subgraphs $K$ in $A$ |
| $\lambda$ | $\mu_0(|\boldsymbol{W}|) - \hat{\mu}(|\boldsymbol{W}|)$ | Spectral gap of weight matrix $\boldsymbol{W}$ |
| $\lambda_{imsg}$ | $\frac{1}{|K|} \sum_{i=1}^{|K|} \left( \mu_0(|\boldsymbol{W}_{K_i}|) - \hat{\mu}(|\boldsymbol{W}_{K_i}|) \right)$ | Iterative mean of spectral gap for set of subgraphs $K$ in $\boldsymbol{W}$ |

## 2.2 The Role of Dynamic Sparse Training in Pruning with Ramanujan

**Limitations of Ramanujan theory in model pruning:** Recently Hoang et al. [7] extended the Ramanujan theory to PaI, achieving notable success in ranking PaI-pruned subnetworks with $\Delta r_{imdb}$. Yet, their reliance on graph spectrum properties introduces pressing challenges. These encompass the absence of a direct pruning strategy, an oversight of weight magnitude, and the inherent non-differentiability of graph spectrum properties. Such limitations substantially restrict the wider applicability of the Ramanujan theory to the straightforward pruning approaches.

**Foundational assumption:** This study is anchored on two pivotal hypothesis. **Firstly**, we posit that graph-based metrics can be strategically used as a sampling criterion, allowing for heuristic optimization and sidestepping issues related to non-differentiability. **Secondly**, we believe that the topology and weights of a sparse subnetwork jointly and complementarily influence performance. To substantiate these hypotheses, we aim to identify a *correlation between a mask's topology, its corresponding weight, and the ensuing performance of the sparse subnetwork*. Ultimately, our goal is to develop an actionable pruning method that can be leveraged to improve PaI as well as PaT.

**Dynamic sparse training as an observational tool:** DST [14, 2, 13, 22, 8] is an efficient sampler that (i) can swiftly navigate the mask-weight space through training dynamics evolution and (ii) is known to specialize at the "good performance" subspace (since DST always evolves to lower the training loss), rather than attempting to fit the full space. This saves the sampling burden considerably. Note that, focusing on the "good performance" subspace suffices for our scenarios since most pruning methods deliver "reasonably good" sparse masks. By using DST as an observational tool, we further provide tantalizing evidence on the natural formation of the graph properties (e.g., Ramanujan characteristic) during training.

## 2.3 Observing Temporal Evolution of Sparse DNN using DST

**Setup:** To generate and evaluate results in this section, we utilize the ITOP framework [13] for its reliable performance in monitoring sparse structures and their evolving graph characteristics. Using magnitude pruning and random growth (initial renewal rate of 50%), we sample model masks and weights every 1500 iterations over 250 epochs. After training these unique sparse models, we assess their graph characteristics to base our subsequent observations. Notably, all our sparse subnetworks maintain a **99% unstructured sparsity** (only 1% trained weights remain non-zero).

### 2.3.1 Correlation between Performance and Sparse Topology

From Table 1, we distill the distinctions between the canonical *Ramanujan bound* $\Delta r$ and its iterative variation $\Delta r_{imdb}$. The correlation between these and performance in image classification across evolving sparse structures is visually captured in Figure 1, with time-sampled data color-coded on a gradient scale. Key takeaways include:

**Effectiveness over time:** Our findings show that the performance of fully fine-tuned sparse DNNs improves over time when sampled during the ITOP process. This reaffirms ITOP's (and DST's overall) capability in pinpointing the "good performance" subspace, as discussed in Section 2.2.

**Contradiction to convention:** The bottom row of Figure 1 displays a narrowing trend for $\Delta r$ with increasing performance, challenging previously held beliefs from [19, 15]. This narrowing suggests that the paths for information flow between layers become progressively more constrained.

**Patterned random-growth:** Contrary to the notion that random growth is entirely unpredictable, Figure 1 indicates structured trends between accuracy and both $\Delta r$ and $\Delta r_{imdb}$. This min-max pattern suggests maximizing or minimizing the expansion property, contingent on the chosen informational boundary metric. Interestingly, the increasing expansion of regular sub-graphs within each layer aligned with the intuitive link between information flow rate and performance.

### 2.3.2 Complementary Relationship between Weight and Topology Graphs

Table 1 itemizes one of our contributions in this paper, namely the proposal for a **weighted spectral gap**, $\lambda$ and $\lambda_{imsg}$ as a comprehensive and reliable performance indicator. Figure 2 presents a heatmap that combines information from both the weight magnitudes and graph topology. This heatmap highlights the region of highly performative models in red, while other regions are shown in blue. By

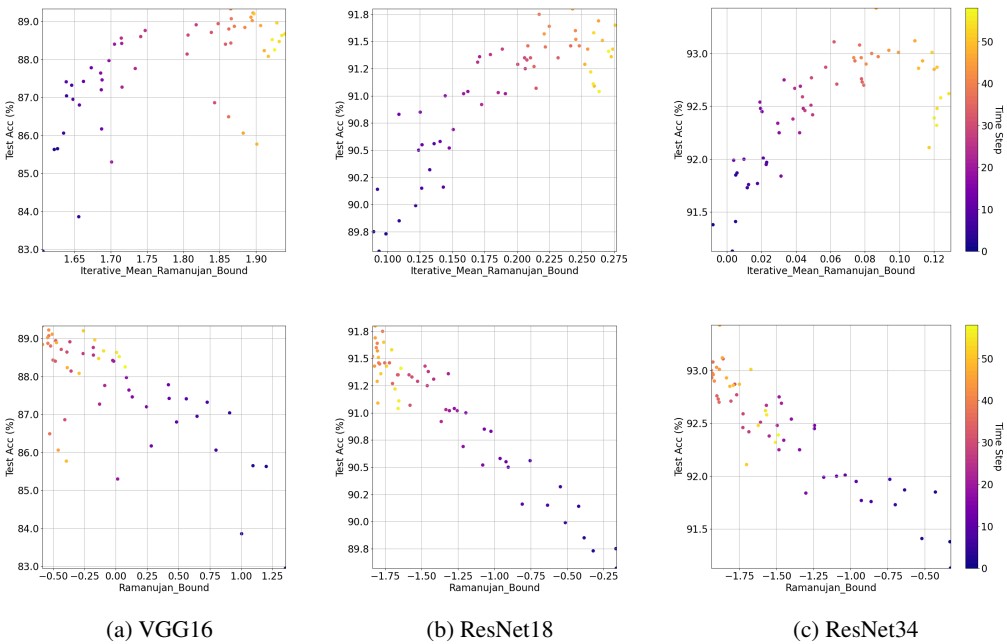

Figure 1: The evolution of $\Delta r_{imdb}$ and $\Delta r$ over time as a performance function using ITOP.

correlating the graph's "unweighte" topology with our proposed weighted counterparts, we make the following observations:

**Pareto curvature:** Figure 2 shows a mutual dependency between the structure of the graph and corresponding weights. In particular, it depicts the Pareto curvature lines between $\Delta r$ and $\lambda$, as well as $\Delta r_{imdb}$ and $\lambda_{imsg}$.

**Area of performance:** We identify the region of highly performative sparse models that lies within a range optimizing the forward expansion characteristic represented by $\Delta r_{imdb}$ while minimizing the average weight magnitude of sub-graphs represented by $\lambda_{imsg}$. Conversely, for the relationship between $\Delta r$ and $\lambda$, improved performance is often associated with narrowing the canonical information flow while increasing the layer-wise weight average.

These observations emphasize the interplay between the structure of the graph and the weights in achieving optimal performance. The analysis in Figure 2 provides valuable insights into the trade-off between graph topology and weight magnitudes, shedding light on the relationship between these factors and performance outcomes. This complex interplay among multiple factors emphasizes the need for unified coordinates to describe the performance relations in sparse DNNs.

## 2.4 Full-Spectrum Coordinate Distance as a Strong, Linear Performance Indicator

In Section 2.3, we identified a robust interconnection between the Ramanujan gap and the weighted spectral gap, from both canonical and iterative angles. Figures 1 and 2 portray the combined influence of sparse topology and weights in optimizing sparse DNN performance, prompting us to investigate the central question: *Is is possible to allow them to co-exist in the same representation*?

To this end, we introduce a novel coordinate system, the **"Full-spectrum"**, integrating $\Delta r$, $\Delta r_{imdb}$, $\lambda$, and $\lambda_{imdb}$ as its four layer-wise axes, given their ties to both weights and topology. Formally, it's represented as $\mathbb{R}^{L \times 4}$, with $L$ denoting the # layers of a model. Navigating this space entails navigating a complex trade-off landscape. We gauge its utility by assessing its linear correlation with overall performance via Pearson correlation. Figures 3 and 4 visualize the efficacy of the "Full-spectrum" in contrast to existing graph metrics. From these, we deduce:

**Consistent linear correlation with performance:** As evidenced by Figure 3 under ITOP's random growth regime, the $\ell_2$-moving distance from a sparse subnetwork's original position within the "Full-spectrum" aligns almost linearly with performance.

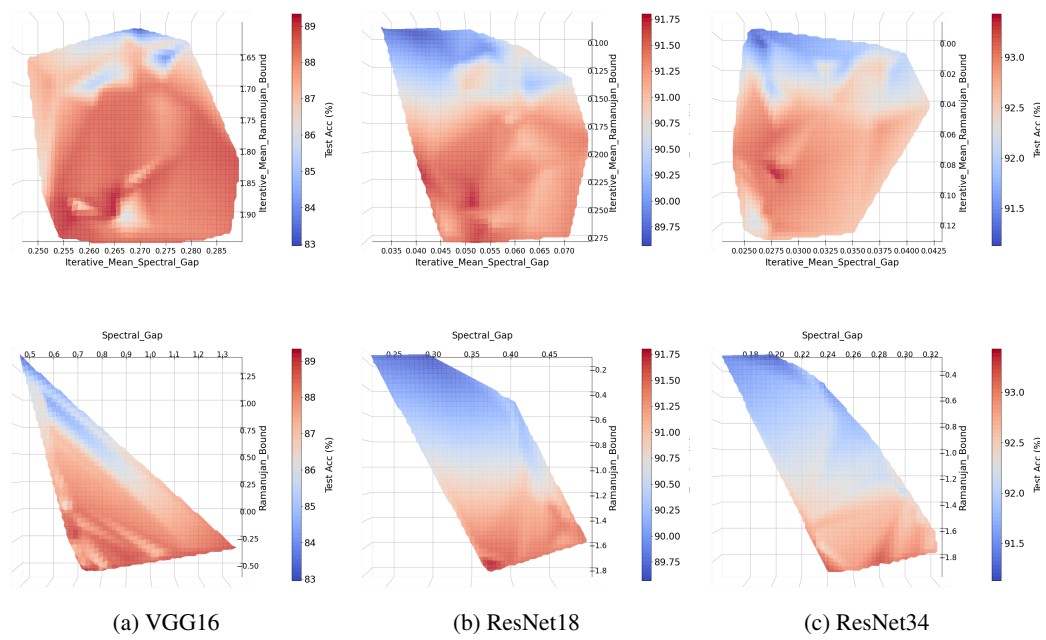

(a) VGG16       (b) ResNet18       (c) ResNet34

Figure 2: Performance landscape between the structure and weights under a graph perspective. In the top panel, we examine the correlation between $\Delta r_{imdb}$ and $\lambda_{imsg}$. In the bottom panel, we explore the correlation between $\Delta r$ and $\lambda$. The landscape represents the classification performance, with gradients coloring from dark blue to dark red, indicating the worst to best performance.

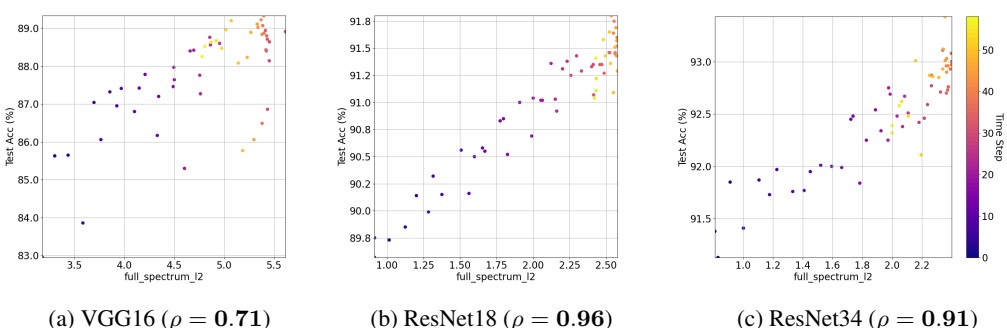

(a) VGG16 ($\rho = \mathbf{0.71}$)      (b) ResNet18 ($\rho = \mathbf{0.96}$)      (c) ResNet34 ($\rho = \mathbf{0.91}$)

Figure 3: Full-spectrum $\ell_2$-distance against classification performance on CIFAR-10 across different models. We denote the associated Pearson correlation ($\rho$) to performance in the parentheses.

**Superiority over current graph metrics:** Figure 4 confirms that, relative to $\Delta r$ and $\lambda$, our "Full-spectrum" perspective offers a more direct linear correlation to performance. This affirms the harmonious co-existence of these metrics within a unified representation, and the potency of this integrated indicator paves the way for its application in tangible pruning scenarios.

## 3 Pruning as Graph Sampling

We now present our actionable pruning methodology as graph sampling, which essentially leverages the proposed $\ell_2$-moving distance metric associated with the "full-spectrum" coordinate, which shows to be a strong, linear performance predictor. Note that our proposed methodology is applicable to any pruning setting since it makes no specific assumption on the weight (e.g., randomly initialized or pre-trained weights) nor the mask (e.g., local layer-wise or global pruning mask). This makes our methodology generic that can be augmented with various "off-the-shelf" pruning methods.

In this section, two specific variants are introduced, one for Pruning at Initialization (PaI), and the other for Pruning after Training (PaT), which requires only a few training iterations.

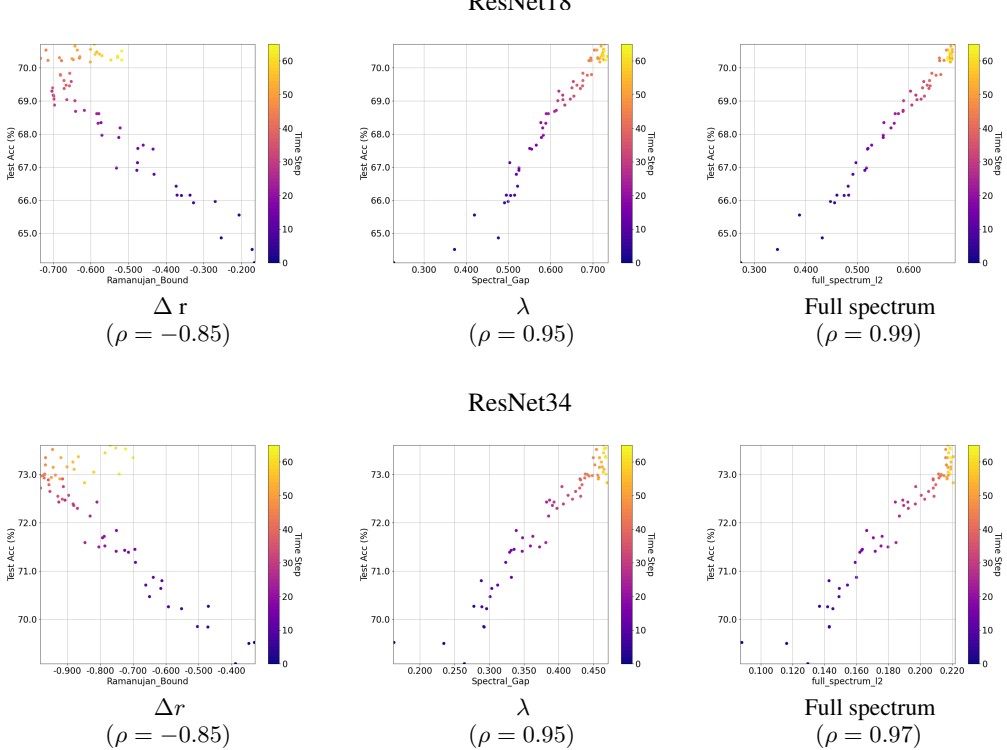

Figure 4: Illustration of the correlation between topology ($\Delta r$), weights ($\lambda$), and the combined "full spectrum" with respect to the classification performance on CIFAR-100 with ResNet18 (top row) and ResNet34 (bottom row). $\rho$ indicates the Pearson correlation.

## 3.1 Pruning at Initialization as Graph Sampling (PAGS)

Pruning at Initialization as Graph Sampling (PAGS) is a lightweight pruning method that does not require any pre-training and aims to maximize the layer-wise full-spectrum $\ell_2$-moving distance. It directly "augments" any existing PaI method, by **oversampling the PaI mask generator** and selecting the top mask that tends to maximize the $\ell_2$-moving distance criteria. In other words, PAGS can be viewed as a "meta-framework" applicable on top of (and to improving) any existing PaI method.

For proxy-driven PaI methods such as [9, 20, 18], we first generate a population of sparsity masks: to create each mask, we simply leverage a randomly selected small minibatch of training data (GenerateMask(.)). For non-proxy-driven PaI like random pruning or ERK [10], the mask population is created by randomly assigning the non-zero values to random weight locations for

---

**Algorithm 1:** Pruning at Initialization as Graph Sampling (PAGS)

1: Initialize the weights $\boldsymbol{\theta}$
2: $n \leftarrow$ population size
3: $k \leftarrow$ mini-batch size
4: $i \leftarrow$ sampling rate
5: $\boldsymbol{A}, \boldsymbol{W} \leftarrow$ GenerateMask(data, $k, \theta$)
6: *// generate sparse topology and weights*
7: **while** $n - 1 \geq 0$ **do**
8:     $\hat{\boldsymbol{A}}, \hat{\boldsymbol{W}} \leftarrow$ GenerateMask(data, $k, \theta$)
9:     *// sampling new mask*
10:    $\boldsymbol{A}, \boldsymbol{W} \leftarrow$ CompareMerge(($\boldsymbol{A}, \boldsymbol{W}$), ($\hat{\boldsymbol{A}}, \hat{\boldsymbol{W}}$))
11:    *// compare and merge if conditions are satisfied*
12:    **if** n $\%$ $i == 0$ and $\boldsymbol{A}, \boldsymbol{W}$ are modified **then**
13:        Save $\boldsymbol{A}, \boldsymbol{W}$
14:    **end if**
15:    $n \leftarrow n - 1$
16: **end while**

---

each mask. From the population, we sample a mask, perform a layer-wise comparison to the current best mask, and update the best in case it improves the $\ell_2$-moving distance (CompareMerge(.)). For a population size of $n$, the process is repeated $n$ times. During these iterations, we periodically save the current best mask at an interval of $i$ steps. Algorithm 1 illustrates this process.

PAGS is **computationally inexpensive**, as each mask sampling requires only a forward pass over a minibatch. In all our experiments, we adopt the default $n = 1,000$, $i = 20$, and minibatch size 128.

Table 2: Results on CIFAR-10 of **PAGS** (with different PaI methods), in comparison to vanilla PaI methods and LTH (as empirical performance "ceiling"). Baseline refers to the PaI-found sparse mask (at initialization) without modification by PAGS. ✓means no pre-training needed (i.e., ideal PaI). ✗, ✗✗, and ✗✗✗, represent low, high, and very high pre-training costs, respectively.

| Method | Baseline acc. % | with PAGS (Best acc %) | with PAGS (Avg. acc %) | No Pretraining needed |
|--------|------|-----------|-----------|-----------|
| | | ResNet18 | | |
| SNIP | 89.54 | 90.17 | $89.80 \pm 0.14$ | ✓ |
| GraSP | 91.39 | **92.01** | $91.68 \pm 0.16$ | ✓ |
| ERK | 88.92 | 90.42 | $91.05 \pm 0.17$ | ✓ |
| Random | 85.43 | 86.03 | $85.72 \pm 0.16$ | ✓ |
| LTH | 91.22 | — | — | ✗✗✗ |
| | | ResNet34 | | |
| SNIP | 91.30 | 91.80 | $91.58 \pm 0.13$ | ✓ |
| GraSP | 91.27 | 91.85 | $91.48 \pm 0.16$ | ✓ |
| ERK | 91.18 | 92.29 | $91.98 \pm 0.13$ | ✓ |
| Random | 88.23 | 88.73 | $88.56 \pm 0.10$ | ✓ |
| LTH | **92.76** | — | — | ✗✗✗ |

## 3.2 Pruning Early as Graph Sampling (PEGS)

To further improve the classification accuracy at reduced compute, we extend the zero-shot PAGS, to present *Pruning Early as Graph Sampling* (**PEGS**), which incurs only a small amount of pre-training cost. In PEGS, we "pre-train" the dense network only for a small number of iterations (by default, for 500 iterations in all our experiments) and then apply PAGS to this lightly trained network. This draws similar inspiration as LTH rewinding [5] or early-bird (EB) ticket [23] in literature.

Compared to the PaI case, we note that it is not as straightforward to directly "sample" masks using LTH/EB or any other PaT method, since generating a different mask from those methods would require re-taking an expensive training process: otherwise, the pruning is deterministic on trained weights and there is no way to inject randomness. Hence, we **do not** treat PEGS as a "meta-framework" on top of PaT methods. **Instead**, we discover that light enough pre-training, followed by cheap PaI-based sampling, can yield sparse masks with comparable quality to the top-performing PaT masks such as LTH, at a significantly cheaper cost. For example, compared to LTH with $M$ rounds of iterative training, $N$ epochs per training round, and $I$ iterations each epoch ($I = 500$ in CIFAR-10): PEGS in the same setting would only cost 500 iterations of pre-training, which is roughly $\mathbf{MN} \times$ **more compute-efficient and faster**. For example, in CIFAR-10 experiments we have $M = 23$ and $N = 250$, hence PEGS would yield $> 5000$ times saving over LTH.

## 4 Experiments

### 4.1 Experiment setup

We leverage four PaI generators to facilitate our experiments (both PAGS and PEGS): • **Random** [10] uniformly prunes every layer with the same pruning ratio assigned globally. Each parameter is randomly assigned a score based on the normal distribution. • **ERK** [3, 14] initializes sparse networks with a *Erdős-Rényi* graph where small layers are usually allocated more budget, while bigger layers are assigned fewer parameter budget. Random pruning is then performed following those layer-wise ratios •**SNIP** [9] for any layer $l$ it issues scores $s_l = |g_l \odot w_l|$ where $g_l$ and $w_l$ are gradients and weights respectively. The weights with the lowest scores after one iteration are pruned before training. •**GraSP** [20] removes weights that impeded gradient flows by computing the Hessian-gradient product $h_l$ and issue scores $s_l = -w \odot h_l$, for a layer $l$.

We also compare with two PaT methods: • **Lottery Ticket Hypothesis** (LTH) [4] iteratively prunes the lowest 20% of weights and rewind the remaining weights to some values in the past. To achieve the desired 99% sparsity, LTH would necessitate 23 rounds of full training and pruning. • **Early Bird (EB)** [23] utilizes one-shot "early pruning", with a "mask-similarity" signal to automatically terminate pre-training as a cost-saving mechanism (typically happening around the first 15%-20% epoches). The original EB was implemented for structured pruning; for a fair comparison, the results shown in Table 3 are our re-implementation of EB onto unstructured pruning.

Table 3: Results on CIFAR-10 of **PEGS** (with light pre-training followed by different PaI methods), in comparison to vanilla PaI methods, LTH and EB (the later two come with much heavier pre-training costs). Baseline refers to the PaI-found sparse mask (after light pre-training) without modification by PEGS. ✗, ✗✗, and ✗✗✗, represent low, high, and very high pre-training costs, respectively.

| Method | Baseline acc. % | with PEGS (Best acc %) | (Avg. acc %) | No pre-training needed |
|---|---|---|---|---|
| | | ResNet18 | | |
| SNIP | 91.05 | **91.53** | 91.22± 0.14 | ✗ |
| GraSP | 89.97 | 91.50 | 91.25± 0.23 | ✗ |
| ERK | 89.87 | 91.31 | 91.05± 0.17 | ✗ |
| Random | 85.37 | 86.03 | 90.13± 0.19 | ✗ |
| EB | 90.10 | — | — | ✗✗ |
| LTH | 91.22 | — | — | ✗✗✗ |
| | | ResNet34 | | |
| SNIP | 92.38 | 92.75 | 92.51± 0.10 | ✗ |
| GraSP | 92.84 | **92.90** | 92.70± 0.40 | ✗ |
| ERK | 91.10 | 92.01 | 91.49± 0.23 | ✗ |
| Random | 87.86 | 88.91 | 88.41±0.23 | ✗ |
| EB | 92.00 | — | — | ✗✗ |
| LTH | 92.76 | — | — | ✗✗✗ |

Table 4: Results on CIFAR-100 on ResNet18 using PAGS/PEGS in comparison to vanilla PaI methods, LTH and EB. Baseline refers to the PaI-found sparse mask. ✗, ✗✗, and ✗✗✗, represent low, high, and very high pre-training costs, respectively. @100 and @500 refer to different "pre-training" iterations using PEGS. @0 means we start from random initialization using PAGS.

| Method | Baseline acc. % | with PAGS/PEGS (Best acc %) | (Avg. acc %) | No pre-training needed |
|---|---|---|---|---|
| | | ResNet18@0 | | |
| SNIP | 64.60 | 65.39 | 64.89± 0.26 | ✓ |
| GraSP | 65.25 | **66.05** | 65.54± 0.21 | ✓ |
| ERK | 64.54 | 64.84 | 64.69 ± 0.12 | ✓ |
| | | ResNet18@100 | | |
| SNIP | 63.22 | 64.92 | 64.34± 0.34 | ✗ |
| GraSP | 63.27 | 65.17 | 64.35± 0.33 | ✗ |
| ERK | 64.06 | 64.82 | 64.41± 0.22 | ✗ |
| | | ResNet18@500 | | |
| SNIP | 62.60 | 64.27 | 63.46± 0.30 | ✗ |
| GraSP | 61.34 | 63.95 | 62.91± 0.59 | ✗ |
| ERK | 64.06 | 65.05 | 64.56± 0.27 | ✗ |
| EB | 62.45 | — | — | ✗✗ |
| LTH | 65.50 | — | — | ✗✗✗ |

Unless otherwise stated, we use a high **target sparsity of** 99% in all our experiments. We demonstrate results on CIFAR-10/ CIFAR-100 in the main text. We use two representative models, Resnet18 and ResNet34, as the main backbones in this section[1]. **Additional training details and results on Tiny-ImangeNet** are deferred to the Supplementary due to the space limit.

## 4.2 Comparison of PAGS and PEGS with Existing PaI methods

Tables 2 and 3 detail our pruning results for CIFAR-10 using PAGS and PEGS, while Tables 4 and 5 do so for CIFAR-100. As PAGS/PEGS operate on a sampling basis, we list both the peak performance of the best mask (Best acc) and the mean performance across all $n$ sampled masks (Avg. acc). The latter is supplemented by the standard deviation. From the presented results, we deduce the following: ① In CIFAR-10/100 trials, our methods consistently surpass the baseline PaI generators in PAGS. A standout instance is the 2.12% advantage over ERK for ResNet18 using PaI, as seen in Table 2. ② With light pre-training, our methods still hold an edge against reference generators. For instance, Tables 4 and 5 indicate about 1% enhancement over the top-performing generator at both the 100 and

---

[1]VGG16 is not included since some PaI methods and EB fail on it at 99% high sparsity ratio.

Table 5: Results on CIFAR-100 on ResNet34 using PAGS/PEGS in comparison to vanilla PaI methods, LTH and EB. Baseline refers to the PaI-found sparse mask. ✗, ✗✗, and ✗✗✗, represent low, high, and very high pre-training costs, respectively. @100 and @500 refer to different "pre-training" iterations using PEGS. @0 means we start from random initialization using PAGS.

| Method | Baseline acc. % | with PAGS/PEGS (Best acc %) | (Avg. acc %) | No pre-training needed |
|--------|--------|--------|--------|--------|
| ResNet34@0 | | | | |
| SNIP | 69.48 | **70.73** | 69.83± 0.27 | ✓ |
| GraSP | 67.88 | 70.59 | 69.64± 0.74 | ✓ |
| ERK | 68.64 | 69.90 | 69.77± 0.11 | ✓ |
| ResNet34@100 | | | | |
| SNIP | 68.04 | 69.41 | 68.64± 0.33 | ✗ |
| GraSP | 62.47 | 66.61 | 64.43 ± 1.03 | ✗ |
| ERK | 68.91 | 69.92 | 69.50 ± 0.16 | ✗ |
| ResNet34@500 | | | | |
| SNIP | 67.41 | 69.23 | 68.53± 0.36 | ✗ |
| GraSP | 67.18 | 68.95 | 68.03 ± 0.41 | ✗ |
| ERK | 68.99 | 69.92 | 69.45± 0.22 | ✗ |
| EB | 65.22 | — | — | ✗✗ |
| LTH | 68.05 | — | — | ✗✗✗ |

500 iteration marks. ③ Interestingly, optimal outcomes typically manifest during the initialization phase, implying that light pre-training doesn't instantaneously refine the weight distribution.

### 4.3 Comparison with (Much More Expensive) LTH and the Early Bird Ticket

We further compare PAGS/PEGS with the two PaT methods, LTH and EB, and note that the latter two are significantly costlier. Overall, for PaI setting (table 2), we intend to include LTH as the "performance ceiling" and show PAGS can get close to or even outperform it often times. For example, Table 2 shows that ResNet18 utilizing GraSP marginally outperforms LTH, without any pre-training.

For PaT setting (table 3, table 4 and table 5), we show that with a small fraction of pre-training costs, our PEGS could **solidly outperform** both LTH and EB, using either SNIP or GRASP mask generators. Interestingly, in ResNet18, even the random ERK mask generator can be turned into LTG-level performance with the aid of PEGS. Our findings suggest that maximizing the full-spectrum's $\ell_2$-moving distance can significantly improve performance by optimizing both the graph's structure and weights. This approach allows us to achieve performance levels comparable to LTH but at a significantly lower cost. By leveraging the insights the Ramanujan perspective provides, we can achieve notable performance improvements across the board of pruning, while incurring minimal computational overheads.

## 5 Conclusion

Recent years have seen the rise of graph theory in analyzing and understanding sparse subnetworks. However, we still lack a crucial understanding of the role parameters play in graph connectivity. To fill this gap, in this paper, we study the weighted spectral gap of Ramanujan structures in sparse neural networks and investigate its correlation with the final performance. By examining the evolution of sparse structures under DST, we identify a strong correlation between Ramanujan bound, weighted spectral gap, and performance. Leveraging these observations, we proposed a new "full-spectrum coordinate" which comprehensively characterizes the complex interplay between various graph characteristics that further leads to actionable pruning methods both at initialization and after light pre-training. This unified perceptive is expected to invoke more future exploration into the complex interplay between topology and weights, not just for sparse NNs but for generic DNNs as well.

## Acknowledgment

Z. Wang is in part supported by NSF Scale-MoDL (award number: 2133861) and the NSF AI Institute for Foundations of Machine Learning (IFML).

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

# A   DNN as bipartite graphs

A bipartite graph is a graph consisting of two distinct sets of vertices $L$ and $R$ that are connected with $E$ edges as $G(L \cup R, E)$. Let $v$ denote the total number of vertices in both sets, the more common way to represent $G$ is as a binary adjacency matrix $\boldsymbol{A} \in \mathbb{R}^{v \times v}$.

A DNN is a series of bipartite graphs, one for each layer, to represent all of its compute graphs. For a convolutional layer with parameter tensor $\boldsymbol{\theta} \in \mathbb{R}^{C_{in} \times C_{out} \times k_{in} \times k_{out}}$, we unfold the dimensions so that $L = C_{in} * k_{in} * k_{out}$ and $R = C_{out}$. For a linear layer with parameter tensor $\boldsymbol{\theta} \in \mathbb{R}^{C_{in} \times C_{out}}$, we can directly adapt its parameters where $L = C_{in}$ and $R = C_{out}$.

An interesting and relevant property of a bipartite graph, is that when each node has the same number of $d$ out-edges, its adjacency matrix $\boldsymbol{A}$ has eigenvalues $\mu(\boldsymbol{A})$ such that $\mu_0 \geq ... \geq \mu_{v-1}$, where $\mu_0$ and $|\mu_{v-1}|$ are equaled to $d$. We define $\hat{\mu}(\boldsymbol{A}) = \max_{|\mu_i| \neq d} |\mu_i|$ as the largest nontrivial eigenvalue. In the context of unstructured pruning, we often find $G$ to be **irregular**, in which case Hoang et al. [7] showed that $d_{avg} \leq |\mu_0| \leq d_{max}$.

# B   Supplementary results for PAGS and PEGS

### B.1   Additional Experiments on CIFAR-10

We expanded our experiments on the CIFAR-10 dataset by utilizing weights pre-trained for 100 iterations with a batch size of 128 per iteration. The CIFAR-10 dataset consists of 50,000 training images and 10,000 testing images, divided into 10 different classes. The results of these experiments are summarized in Table 6.

We observed performance improvement relative to baseline. However, compared to other modes of pre-training for CIFAR-10, certain PaI generators exhibited higher-than-expected standard deviation and lower average performance, indicating some instability in generating sparse structures. Specifically, we observed this trend with GraSP in ResNet18 and SNIP in ResNet34.

### B.2   Additional Experiments on Tiny-Imagenet

We expanded our experiments on the Tiny-Imagenet dataset by utilizing weights pre-trained for 100 iterations with a batch size of 128 per iteration. The Tiny-Imagenet dataset consists of 100,000 images, divided into 200 different classes. The results of these experiments are summarized in table 7 and table 8.

# C   Limitations and Societal Impacts

This work studies the effect of weights under the Ramanujan settings through observation using ITOP. By gaining insights from these observations, we empirically improve the performance of pruning methods using PAGS. We do not expect any negative societal impact from this work.

Table 6: Results on CIFAR10 on ResNet18 and Resnet34 using PEGS in comparison to vanilla PaI methods, LTH and EB. Baseline refers to the PaI-found sparse mask. ✗, ✗✗, and ✗✗✗, represent low, high, and very high pre-training costs, respectively. @100 refers to weight pretraining at 100 iterations with batch size 128.

| Method | Baseline acc. % | with PEGS (Best acc %) | with PEGS (Avg. acc %) | No pre-training needed |
|--------|------|------|------|------|
| ResNet18@100 | | | | |
| SNIP | 90.39 | **91.30** | 90.06± 0.17 | ✓ |
| GraSP | 86.09 | 91.11 | 80.38± 3.14 | ✓ |
| ERK | 90.07 | 90.41 | 90.11± 0.22 | ✓ |
| EB | 90.10 | — | — | ✗✗ |
| LTH | 91.22 | — | — | ✗✗✗ |
| ResNet34@100 | | | | |
| SNIP | 92.91 | **93.22** | 90.01± 1.70 | ✓ |
| GraSP | 92.66 | 93.11 | 92.91± 0.11 | ✓ |
| ERK | 92.04 | 92.19 | 91.88± 0.18 | ✓ |
| EB | 92.00 | — | — | ✗✗ |
| LTH | 92.76 | — | — | ✗✗✗ |

Table 7: Results on Tiny-ImageNet on ResNet18 using PAGS/PEGS in comparison to vanilla PaI methods, LTH and EB. Baseline refers to the PaI-found sparse mask. ✗, ✗✗, and ✗✗✗, represent low, high, and very high pre-training costs, respectively. @100 and @500 refer to different "pre-training" iterations using PEGS.

| Method | Baseline acc. % | with PAGS/PEGS (Best acc %) | with PAGS/PEGS (Avg. acc %) | No pre-training needed |
|--------|------|------|------|------|
| ResNet18@100 | | | | |
| SNIP | 47.13 | 49.00 | 48.45 ± 0.44 | ✗ |
| GraSP | 48.31 | 49.43 | 48.82 ± 0.32 | ✗ |
| ResNet18@500 | | | | |
| SNIP | 45.83 | 47.83 | 47.11 ±0.42 | ✗ |
| GraSP | 48.89 | 50.26 | 49.48 ± 0.29 | ✗ |
| EB | 47.43 | — | — | ✗✗ |
| LTH | 49.81 | — | — | ✗✗✗ |

Table 8: Results on Tiny-ImageNet on ResNet34 using PAGS/PEGS in comparison to vanilla PaI methods, LTH and EB. Baseline refers to the PaI-found sparse mask. ✗, ✗✗, and ✗✗✗, represent low, high, and very high pre-training costs, respectively. @100 and @500 refer to different "pre-training" iterations using PEGS.

| Method | Baseline acc. % | with PAGS/PEGS (Best acc %) | with PAGS/PEGS (Avg. acc %) | No pre-training needed |
|--------|------|------|------|------|
| ResNet34@100 | | | | |
| SNIP | 53.07 | 54.91 | 53.65 ± 0.4 | ✗ |
| GraSP | 52.71 | 53.71 | 52.80 ± 0.41 | ✗ |
| ResNet34@500 | | | | |
| SNIP | 53.57 | 54.65 | 54.16 ± 0.25 | ✗ |
| GraSP | 53.83 | 55.48 | 54.84 ± 0.35 | ✗ |
| EB | 53.40 | — | — | ✗✗ |
| LTH | 54.00 | — | — | ✗✗✗ |

