# Don't Just Prune by Magnitude! Your Mask Topology is Another Secret Weapon

## A  Supplementary results for PAGS and PEGS

### A.1  Additional Experiments on CIFAR10

We expanded our experiments on the CIFAR10 dataset by utilizing weights pretrained for 100 iterations with a batch size of 128 per iteration. The CIFAR10 dataset consists of 50,000 training images and 10,000 testing images, divided into 10 different classes. The results of these experiments are summarized in Table 1.

We observed performance improvement relative to baseline. However, compared to other modes of pretraining for CIFAR10, certain PaI generators exhibited higher-than-expected standard deviation and lower average performance, indicating some instability in generating sparse structures. Specifically, we observed this trend with GraSP in ResNet18 and SNIP in ResNet34.

Table 1: Results on CIFAR10 on ResNet18 and Resnet34 using PEGS in comparison to vanilla PaI methods, LTH and EB. Baseline refers to the PaI-found sparse mask. ✗, ✗✗, and ✗✗✗, represent low, high, and very high pre-training costs, respectively. @100 refers to weight pretraining at 100 iterations with batch size 128.

| Method | Baseline acc. % | with PEGS (Best acc %) | with PEGS (Avg. acc %) | No pre-training needed |
|---|---|---|---|---|
| | | ResNet18@100 | | |
| SNIP | 90.39 | **91.30** | 90.06± 0.17 | ✓ |
| GraSP | 86.09 | 91.11 | 80.38± 3.14 | ✓ |
| ERK | 90.07 | 90.41 | 90.11± 0.22 | ✓ |
| EB | 90.10 | — | — | ✗✗ |
| LTH | 91.22 | — | — | ✗✗✗ |
| | | ResNet34@100 | | |
| SNIP | 92.91 | **93.22** | 90.01± 1.70 | ✓ |
| GraSP | 92.66 | 93.11 | 92.91± 0.11 | ✓ |
| ERK | 92.04 | 92.19 | 91.88± 0.18 | ✓ |
| EB | 92.00 | — | — | ✗✗ |
| LTH | 92.76 | — | — | ✗✗✗ |

### A.2  Experiments and Observations on CIFAR100

We conducted experiments on the CIFAR100 dataset, which consists of 50,000 training images and 10,000 testing images across 100 different classes. The experiments were performed on ResNet18 and ResNet34 models under three different settings: using random weights, using weights pretrained for 100 iterations, and using weights pretrained for 500 iterations. The batch size per iteration was set

to 128. The results for ResNet18 are reported in Table 2, and the results for ResNet34 are reported in Table 3.

Overall, we observed that light-pretrained weights typically performed worse than random initialization for the CIFAR100 dataset.

Additionally, we performed a similar analysis using the ITOP framework on CIFAR100. Figures 1 and 2 illustrate the correlations between different metrics (topology and weights) and performance, as well as our proposed "Full-spectrum" $\ell_2$-distance metric. It is noteworthy that the Pearson's correlation increases progressively from topology to weights to $\ell_2$-distance.

Table 2: Results on CIFAR100 on ResNet18 using PAGS/PEGS in comparison to vanilla PaI methods, LTH and EB. Baseline refers to the PaI-found sparse mask. ✗, ✗✗, and ✗✗✗, represent low, high, and very high pre-training costs, respectively. @100 and @500 refer to different "pre-training" iterations using PEGS. @0 means we start from random initialization using PAGS.

| Method | Baseline acc. % | with PAGS/PEGS (Best acc %) | with PAGS/PEGS (Avg. acc %) | No pre-training needed |
|---|---|---|---|---|
| | | ResNet18@0 | | |
| SNIP | 64.60 | 65.39 | 64.89± 0.26 | ✓ |
| GraSP | 65.25 | **66.05** | 65.54± 0.21 | ✓ |
| ERK | 64.54 | 64.84 | 64.69 ± 0.12 | ✓ |
| | | ResNet18@100 | | |
| SNIP | 63.22 | 64.92 | 64.34± 0.34 | ✗ |
| GraSP | 63.27 | 65.17 | 64.35± 0.33 | ✗ |
| ERK | 64.06 | 64.82 | 64.41± 0.22 | ✗ |
| | | ResNet18@500 | | |
| SNIP | 62.60 | 64.27 | 63.46± 0.30 | ✗ |
| GraSP | 61.34 | 63.95 | 62.91± 0.59 | ✗ |
| ERK | 64.06 | 65.05 | 64.56± 0.27 | ✗ |
| EB | 62.45 | — | — | ✗✗ |
| LTH | 65.50 | — | — | ✗✗✗ |

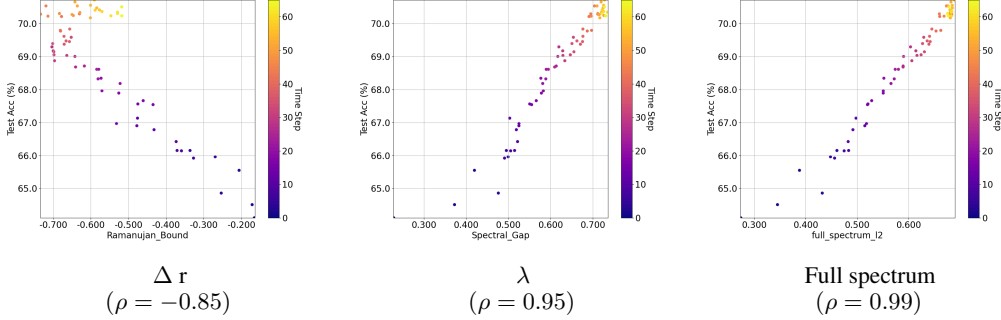

| $\Delta$ r | $\lambda$ | Full spectrum |
|---|---|---|
| ($\rho = -0.85$) | ($\rho = 0.95$) | ($\rho = 0.99$) |

Figure 1: this figure illustrates the correlation between topology ($\Delta r$), weights ($\lambda$), and the combined "full spectrum" with respect to CIFAR-100's classification performance for ResNet18 model. $\rho$ indicates the Pearson correlation.

# B    Limitations and Societal Impacts

This work study the effect of weights under the Ramanujan settings through observation using ITOP. By gaining insights from these observations, we empirically improve the performance of pruning methods using PAGS. We do not expect any negative societal impact from this work.

# References

Table 3: Results on CIFAR100 on ResNet34 using PAGS/PEGS in comparison to vanilla PaI methods, LTH and EB. Baseline refers to the PaI-found sparse mask. ✗, ✗✗, and ✗✗✗, represent low, high, and very high pre-training costs, respectively. @100 and @500 refer to different "pre-training" iterations using PEGS. @0 means we start from random initialization using PAGS.

| Method | Baseline acc. % | with PAGS/PEGS (Best acc %) | with PAGS/PEGS (Avg. acc %) | No pre-training needed |
|---|---|---|---|---|
| | | ResNet34@0 | | |
| SNIP | 69.48 | **70.73** | 69.83± 0.27 | ✓ |
| GraSP | 67.88 | 70.59 | 69.64± 0.74 | ✓ |
| ERK | 68.64 | 69.90 | 69.77± 0.11 | ✓ |
| | | ResNet34@100 | | |
| SNIP | 68.04 | 69.41 | 68.64± 0.33 | ✗ |
| GraSP | 62.47 | 66.61 | 64.43 ± 1.03 | ✗ |
| ERK | 68.91 | 69.92 | 69.50 ± 0.16 | ✗ |
| | | ResNet34@500 | | |
| SNIP | 67.41 | 69.23 | 68.53± 0.36 | ✗ |
| GraSP | 67.18 | 68.95 | 68.03 ± 0.41 | ✗ |
| ERK | 68.99 | 69.92 | 69.45± 0.22 | ✗ |
| EB | 65.22 | — | — | ✗✗ |
| LTH | 68.05 | — | — | ✗✗✗ |

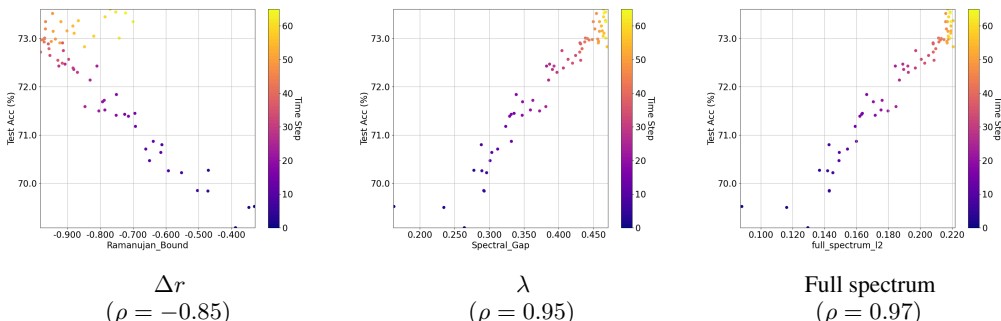

$$\Delta r \qquad\qquad \lambda \qquad\qquad \text{Full spectrum}$$
$$(\rho = -0.85) \qquad (\rho = 0.95) \qquad (\rho = 0.97)$$

Figure 2: this figure illustrates the correlation between topology ($\Delta r$), weights ($\lambda$), and the combined "full spectrum" with respect to CIFAR-100's classification performance for ResNet34 model. $\rho$ indicates the Pearson correlation.