# OpenReview forum: "Don’t just prune by magnitude! Your mask topology is a secret weapon"
_NeurIPS.cc/2023/Conference — NeurIPS 2023 poster_

### Official Review · Reviewer_62wf · 2023-07-04

**Soundness:** 2 fair
**Presentation:** 2 fair
**Contribution:** 2 fair
**Rating:** 5
**Confidence:** 4

**Summary:**

The paper suggests two pruning strategies: pruning at initialization as graph sampling (PAGS) and pruning early as graph sampling (PEGS). The PAGS approach is one of those that doesn't call for any pre-training dietary intake. The mask is oversampled each time, and the masks with the greatest l2 movement distance are chosen. With the PEGS approach, which is a slight enlargement of PAGS, the dense network undergoes very little pre-training. The article experiment performs SOTA on CIFAR10.

**Strengths:**

* The context of the paper is very logical.
* This paper is rich in theoretical analysis.

**Weaknesses:**

* The actual method suggested is relatively naive, and too much length in the paper is devoted to analysis and introduction.

* The experimental part only contains the CIFAR-10/100 dataset, and such a small number of experiments cannot accurately reflect the effectiveness of the method. It is highly recommended that the author increase the validity experiments such as ImagNet.

* "only 500 iteration" does not support the statement of "a few iteration" on small datasets like CIFAR.

**Questions:**

Please see the weakness part.

**Limitations:**

No limitations discussed.

---

> ### Author Rebuttal · Authors · 2023-08-09
>
> 1) The actual method suggested is relatively naive, and too much length in the paper is devoted to analysis and introduction.
>
> > We thank the reviewer for their comment. We understand that both PAGS and PEGS can be considered as simple greedy algorithms. However, we consider this to be an advantage of our method to apply these techniques in a "plug-and-play" way on top of existing SOTA low-cost pruning methods as discussed. Additionally, we would like to highlight that these methods are the outcome of our important and novel observations. In particular, we first showed the directed nature of  random evolution in sparse topology which follows the Ramanujan perspective. We further  illustrated the existence of an almost linear correlation between performance and the combined coordinates of the Ramanujan and weighted spectral gap. And finally, we derived simple methods that reliably sample performative masks at initialization, stemming from these observations.
>
> > We apologize for the insufficient readability of our paper. In our next revision, we'll simplify both our observations and the methods as suggested by the reviewer (while incorporating all the readability improvement suggestions of other reviewers as well). Additionally, in the global rebuttal, we have summarized the major take-away from each section which we will follow in the revised draft to improve readability.
>
>
> 2) The experimental part only contains the CIFAR-10/100 dataset, and such a small number of experiments cannot accurately reflect the effectiveness of the method. It is highly recommended that the author increase the validity experiments such as ImageNet.
>
> > Please see our global responses
>
> 3) "only 500 iteration" does not support the statement of "a few iteration" on small datasets like CIFAR.
>
> > We appreciate the reviewer's feedback. We concur that our choice of 500 iterations with a batch size of 128 might raise questions; however, this equates to approximately one epoch on CIFAR-10. The rationale behind this choice was to align with the default pre-training parameters set by Lottery Ticket Hypothesis (LTH) in OpenLTH repository [1], ensuring a fair comparison of results. We will make certain to clarify this aspect in our revised version.
>
> [1]https://github.com/facebookresearch/open_lth

---

> > ### Comment · Reviewer_62wf · 2023-08-15
> > **Thanks**
> >
> > Thanks for the rebuttal. In summary, I believe this paper makes theoretical contributions, but the novelty of the method is insufficient, and the experiments are weak (the added experiments are still just on tiny-imagenet, far from large-scale datasets and it is common sense in the community for evaluating pruning algorithms on ImageNet). Lastly, I hold concerns about the practical value of this research direction, as the performance of PAI is, to my knowledge, far inferior to sparse training methods [1][2], and neither requires the overhead of pre-training. Therefore, I can't recommend the current version to be accepted by top-tier conferences like NeurIPS.
> >
> > [1] Rigging the lottery: Making all tickets winners. In ICML, 2020.
> > [2] Sparse Training via Boosting Pruning Plasticity with Neuroregeneration. In NeurIPs, 2021.

---

> > > ### Author Response · Authors · 2023-08-15
> > > **Thank you, but we disagree.**
> > >
> > > We appreciate the reviewer's feedback and their commendation of our theoretical contributions. But we disagree with both of your accusations, with respect.
> > >
> > > (1)
> > > In response to the comment on our experimental scale, we’d like to highlight an oral paper from ICLR 2023 [1] and a main direct inspiration for our idea. This paper effectively used only CIFAR to demonstrate the potential of Ramanujan’s graph, and that doesn't compromise its shining contribution unanimously appreciated by its reivewers. We are happy to list more examples at all recent "top conferences" in this flavor, if you would like to see more.
> > >
> > > Once again, we would like to emphasize the larger context of our paper, which is to showcase how random masks, when viewed through the Ramanujan perspective, can offer greater sparse masks that used to be only attainable by more expensive means. In this perspective, our experiments are designed to validate these observations and our theoretical contributions, rather than to establish one more performative ad-hoc pruning.
> > >
> > > (2)
> > > On the topic of the practical value of our research direction, the author team consists of very seasoned experts with PAI and dynamic sparse training (DST), publishing extensively in both ends and beyond. We are very familiar with the performance disparity between PaI and DST (esp. RigL). We refer to the article [2] for an excellent detailed perspective on (1) why PaI is meaningful; and (2) why DST and PAI should be addressed distinctively.
> > >
> > > To put it short: DST methods including RigL (a) need change masks through training which has unrealistic overhead in practice while training with one fixed sparse mask is much more straightforward in implementation; (b) DST typically required extended training epochs while PaI trains the same length of epochs as standard dense training; (c) PaI is considered as an important theoretical & insight tool to study "optimal sparse network design" that is independent of (prior to) training; while DST offers effective ad-hoc in-training algorithms.
> > >
> > >
> > > While we take reviewers’ concerns seriously and aim to address them respectfully, we kindly request reviewers extend the same respect to the research field as a whole (PaI, and theory-driven pruning).
> > > - It is certainly important to avoid unfairly dismissing the PaI field which establishes important insights for optimal sparse network design (training-independent).
> > > - It is equally important to NOT blindly assess/accuse papers with theoretical focus, by whether they perform “large-scale experiment” as this could easily fall outside their specific goal and literature context.
> > >
> > > [1] D. N. Hoang, S. Liu, R. Marculescu, and Z. Wang. Revisiting Pruning at Initialization Through the Lens of Ramanujan Graph. In The Eleventh International Conference on Learning Representations, 2023.
> > >
> > > [2] https://arxiv.org/abs/2302.02596

---

> > > > ### Author Response · Authors · 2023-08-18
> > > > **Following up on our discussion**
> > > >
> > > > Dear Reviewer 62wf,
> > > >
> > > > As the discussion period concludes, we aim to confirm that we have comprehensively addressed your concerns. Through our recent discussion, we hope to have highlighted that our experiments align with contemporary research of a similar focus [1]. Moreover, we believe we have presented a case for both PaI and DST as robust and worthy of investigation. Should there be any lingering uncertainties or concerns about our work, we invite you to relay them so we might offer further insights. We genuinely hope our exchanges have underscored our paper's important contributions, and we eagerly anticipate your reconsideration.
> > > >
> > > > Warm regards,
> > > >
> > > > Authors
> > > >
> > > > [1] D. N. Hoang, S. Liu, R. Marculescu, and Z. Wang. Revisiting Pruning at Initialization Through the Lens of Ramanujan Graph. In The Eleventh International Conference on Learning Representations, 2023.

---

### Official Review · Reviewer_QXrC · 2023-07-04

**Soundness:** 3 good
**Presentation:** 3 good
**Contribution:** 3 good
**Rating:** 7
**Confidence:** 3

**Summary:**

This paper proposes a new method for pruning deep neural networks that optimizes both the network's structure and weights. The experimental results show that it can be naturally applied as a “zero-shot" PaI method and outperforms existing PaI methods with large margins. Alternatively, it can be applied after light pre-training to get LTH-comparable performance.

**Strengths:**

The proposed approach to deep network pruning based on Ramanujan graphs is a technically innovative and interesting method that optimizes both the graph's structure and weights, achieving notable performance improvements across various pruning methods while incurring minimal computational overheads. As connecting graph theory to sparse network architecture is a recently emergent idea, this paper fills in a timely and important research gap.

The authors leveraged DST as an organic tool to sample sparse masks and observe its co-evolution with weights. The research observation sheds light on the relationship between the connectivity of a neural network's architecture as a graph and the network's performance, providing a complementary perspective to understand the effectiveness of different pruning methods.

The algorithm novelty lies in the use of a new "full-spectrum coordinate" that characterizes a sparse neural network's promise, consisting of the classical Ramanujan's gap, the proposed weighted spectral gap, and the constituent nested regular graphs within.

The experimental findings suggest that maximizing the full-spectrum's l2-moving distance can significantly improve performance by optimizing both the graph's structure and weights. The empirical evidence also shows that even the random ERK mask generator can be turned into LTH-level performance with the aid of PEGS.


**Weaknesses:**

Despite the appeal of theory, I don’t think the current experiments are sound enough. In general, all pruning methods in-use tend to perform similarly on CIFAR-10/100. As seen in Tables 1 and 2, different methods’ numbers are often within each other’s error bar, making it hard to judge whether one “outperforms” others with true statistical significance or just by chance/ random noise.

Noteworthily, while the authors highlighted their success that “ResNet18 utilizing GraSP (+ PAGS) even marginally outperforms LTH, without any pretraining”, the GRASP baseline in Table 1 (top) and Table 2 (bottom) already outperformed LTH, hence disapproving LTH as “empirical performance ceiling” as authors assumed (or otherwise, implying there exists an unfair comparison among baselines).

I also don’t feel the “Full-Spectrum Coordinate” is convincing nor sound enough. An ad-hoc construction by authors, it has two “topology-only” coordinates and two other “weight-topology” coordinates. So at least, the four two groups of coordinates are redundant, one dependent of the other. Why one cannot expect a better linear predictor via more “orthogonalized” coordinates, such as two “topology-only” and two “weight-only” coordinates combined?

Lastly, the experiment part contains many hyperparameters that were barely discussed or justified. For example, “I = 50”, “N = 250”, etc. How are they chosen and are they sensitive?



**Questions:**

This is overall an interesting and theoretically sound paper, but several important concerns remain as discussed in “weakness”. Please address each of them.

---

> ### Author Rebuttal · Authors · 2023-08-09
>
> 1) Despite the appeal of theory, I don’t think the current experiments are sound enough. In general, all pruning methods in-use tend to perform similarly on CIFAR-10/100. As seen in Tables 1 and 2, different methods’ numbers are often within each other’s error bar, making it hard to judge whether one “outperforms” others with true statistical significance or just by chance/ random noise.
>
> > Please refer to our global responses.
>
>
> 2) Noteworthily, while the authors highlighted their success that “ResNet18 utilizing GraSP (+ PAGS) even marginally outperforms LTH, without any pretraining”, the GRASP baseline in Table 1 (top) and Table 2 (bottom) already outperformed LTH, hence disapproving LTH as “empirical performance ceiling” as authors assumed (or otherwise, implying there exists an unfair comparison among baselines).
>
> > In smaller datasets such as CIFAR, we observe PaI methods can yield competitive results, even compared to LTH. However, this phenomenon quickly diminishes with larger datasets like Tiny-ImageNet (please refer to global response for results). We will revise the draft to update based on this observation.
>
> 3) I also don’t feel the “Full-Spectrum Coordinate” is convincing nor sound enough. An ad-hoc construction by authors, it has two “topology-only” coordinates and two other “weight-topology” coordinates. So at least, the four two groups of coordinates are redundant, one dependent on the other. Why one cannot expect a better linear predictor via more “orthogonalized” coordinates, such as two “topology-only” and two “weight-only” coordinates combined?
>
> > We thank the reviewer for their criticism. From our understanding, the reviewer is questioning our decision to use four separate criteria (two for weights and two for topology) instead of just two (one for topology and one for weights), as they believe the two metrics for either weights or topology are fundamentally interdependent.
> > We argue that these metrics should not be combined as they measure different forms of information flow and are not fundamentally dependent on one another. In support of our decision, we have cited prior work such as [1] that clearly demonstrates the differences in effectiveness in using the Ramanujan gap to predict performance between the iterative and non-iterative versions. Similarly, we have described their distinctions between lines 155-169.
> > In summary, the non-trivial eigenvalue gauges the general information boundary of the entire compute graph, without considering its regular sub-graphs, and can sometimes be overly strict for irregular graphs. On the other hand, the iterative version evaluates the average information boundary of sub-graphs within a larger structure, a criteria shown to be more effective by [1]. Our decision to use both stems from our observation of potential trends exhibited by both measures in random sparse networks in DST. These differences in what they measure justify our approach of using separate criteria. We will ensure that our revision includes a clear explanation to avoid any confusion.
>
>
>
>
> 4) Lastly, the experiment part contains many hyperparameters that were barely discussed or justified. For example, “I = 50”, “N = 250”, etc. How are they chosen and are they sensitive?
>
> > We selected these parameters to match LTH's default parameters in the OpenLTH repository[2]. This alignment was intentional to ensure a fair and consistent comparison with the benchmark. We will emphasize this point in the experimental settings section in future revision.
>
> [1] D. N. Hoang, S. Liu, R. Marculescu, and Z. Wang. Revisiting Pruning at Initialization Through 355 the Lens of Ramanujan Graph. In The Eleventh International Conference on Learning Representations, 2023.
>
> [2]https://github.com/facebookresearch/open_lth

---

> > ### Comment · Reviewer_QXrC · 2023-08-15
> >
> > Thanks for the detailed rebuttal from authors, my concerns are properly addressed.

---

> > > ### Author Response · Authors · 2023-08-15
> > > **Thank You!**
> > >
> > > We'd appreciate the reviewer for positively commenting our work and raising the score to 7! This is a great encouragement and we'll improve our work per discussion.

---

### Official Review · Reviewer_9NeG · 2023-07-06

**Soundness:** 3 good
**Presentation:** 2 fair
**Contribution:** 3 good
**Rating:** 7
**Confidence:** 4

**Summary:**

This paper proposes a new approach to deep network pruning based on graph theory, specifically the use of Ramanujan graphs. The authors argue that their approach, which focuses on optimizing both the graph's structure and weights, can achieve notable performance improvements across various pruning methods while incurring minimal computational overheads. Empirical results demonstrate that their approach outperforms existing methods such as LTH and EB, even with a small fraction of pre-training costs.

**Strengths:**

-	The paper proposes a new approach to deep network pruning based on Ramanujan graphs, which are highly sparse yet highly connected graphs that align with the primary objective of pruning. Although prior arts have connected Ramanujan graphs with pruning/PaI quality assessment, this is first work deriving an actionable pruning method based on such.

-	The approach focuses on optimizing both the graph's structure and weights, which can achieve notable performance improvements across various pruning methods while incurring minimal computational overheads.

-	Empirical results demonstrate that the proposed approach outperforms existing methods such as LTH and EB, even with a small fraction of pre-training costs. It also significantly improve the previous random sampling-based pruning, e.g. ERK.

-	The paper also sheds light on the relationship between the connectivity of a neural network's architecture as a graph and the network's performance, providing a complementary perspective to understand the effectiveness of different pruning methods.




**Weaknesses:**

I have two major critiques: (1) poor readability, and (2) unimpressive, sometimes confusing experiments.

Readability: This paper is, at very least, quite hard to follow. That is partially owing to the dense content – the authors tried to pack too many observational stuff in one submission (sections 2-4, before finally reaching their “actionable pruning” results in section 5). That is along with a lot of self-defined jargons and unnecessary notations. I suggest merging sections 2-4 into two sections, one on observation and the other on pruning method, both being significantly shorter in length.

Experiments: there are several presented results confusing me.

-	Why VGG shows much poorer linear correlation in Figures 1-4, compared to ResNets? Also the authors didn’t report VGG results in Section 5. I wonder: is the proposed method only specifically effective on ResNets?

-	I am not sure I understand how to read Figure 3. There is no detailed analysis in main text either.

-	In Table 2, the improvements are veryminor, usually within error bar. I am not convinced the improvement of PEGS is evident. Can you show more dataset results like ImageNet?

-	In Tables 1 and 2: do you suggest it’s fairer to compare baseline acc with “Best acc”, or “Average acc”?


**Questions:**

Please check questions enlisted above.

**Limitations:**

The authors have not explicitly discussed any limitation.

---

> ### Author Rebuttal · Authors · 2023-08-09
>
> 1) I have two major critiques: (1) poor readability, and (2) unimpressive, sometimes confusing experiments.
> * Poor readability
>
> * Experiments: there are several presented results confusing me. Why VGG shows much poorer linear correlation in Figures 1-4, compared to ResNets? Also the authors didn’t report VGG results in Section 5. I wonder: is the proposed method only specifically effective on ResNets?
>
> > For issue with readability please refer to our global responses.
>
> > For experiments, the reason why VGG exhibited a weaker correlation compared to ResNet lies in its unstable nature, stemming from the absence of skip-connections. In our Dynamic Sparse Training (DST)  settings, we perform unstructured pruning on all the models at 1% of their original parameter size using a random-growth policy. For VGG, lacking skip connections, this can lead to poorly aligned layer-wise sparsity masks, potentially affecting information flow and thus disrupting performance. Since Figures [1-4] depict the performance of masks we sampled over time, VGG is more prone to producing outliers with infeasible masks, thereby diminishing its correlation to performance.
> We chose not to include VGG methods in Section 5, as indicated in our footnote, due to the propensity for PaI methods to fail in generating feasible masks for sampling. Our sampling approach, which relies on PaI for mask proposal, is notably more reliable for networks with some form of residual skip-connection. We recognize this constraint and will ensure to address it in the limitations section of our paper.
>
>
> 2) I am not sure I understand how to read Figure 3. There is no detailed analysis in main text either.
>
> > Figure 3 provides visual representation of three essential metrics: the x-axis shows the Ramanujan-bound (topology), the y-axis depicts the spectral gap (weights), and the colored landscape illustrates performance (accuracy). The objective of this figure is to swiftly convey the inherent relationship between topology, weights, and performance. Notably, the region of peak performance (highlighted in bright red) is concentrated in a specific area. This distinct region is the target we seek to optimize in our sampling method. We highlight this interdependency between the metrics in Observation 1. In the next revision, we will ensure to clarify the meaning and observations related to Figure 3.
>
> 3) In Table 2, the improvements are very minor, usually within the error bar. I am not convinced the improvement of PEGS is evident. Can you show more dataset results like ImageNet?
>
> > Please refer to our global responses.
>
>
> 4) In Tables 1 and 2: do you suggest it’s fairer to compare baseline acc with “Best acc”, or “Average acc”?
>
> > We utilize average accuracy to show  confidence in our sampling process, showing that the average performance over the entire sampling window exceeds that of the original mask. However, we find it reasonable to compare the best sampled results with baseline performance, as the average calculation includes both fully optimized and partially optimized masks.

---

> > ### Comment · Reviewer_9NeG · 2023-08-15
> >
> > Thanks for the author's response. I've reviewed the questions from other reviewers and the author's responses. I believe the author has adequately addressed the concerns of both myself and other reviewers.

---

### Official Review · Reviewer_i7oA · 2023-07-07

**Soundness:** 3 good
**Presentation:** 1 poor
**Contribution:** 2 fair
**Rating:** 3
**Confidence:** 2

**Summary:**

This work studies network pruning in a way that takes into account both the graph topology and values of weights. This connects purely magnitude-based and purely graph-based pruning methods. In particular, the authors find correlations between performance of masks and 4 properties derived per layer of the neural network that account for graph structure and weight magnitude. The authors then use these properties to propose a new method that augments existing pruning at initialization methods.

**Strengths:**

1. Sampling after a light pretraining is a good, efficient approach.
2. Proposed PAGS method is simple in a good sense, and can be used with any other pruning at initialization method.
3. Empirical contribution of measurements of various network properties during ITOP.


**Weaknesses:**

1. Page 5's "physics intuition" is hard to understand and unrigorous, and does not seem to do much to alleviate the mentioned "lack of theory" from the Pal et al. approach.
2. [1] also uses weighted graph regularization to sparsify neural networks weights.
3. I generally find the paper hard to read. It seems like there is a lot of content and jargon / abbreviations introduced very rapidly.
4. Use of weighted spectral gap is not so novel.
5. Experimental improvements are quite modest in Tables 1 and 2.
6. It would be interesting to more directly optimize for the 4 properties of the coordinate system, instead of simplying sampling and choosing masks that improve on those properties.

[1] Tam and Dunson. Fiedler Regularization: Learning Neural Networks with Graph Sparsity. ICML 2020.


**Questions:**

Page 3, the use of "tantalizing" is hard to understand for me here. What did you mean by this?


**Limitations:**

There is a limitations section in Appendix B, but it does not actually discuss any limitations.

---

> ### Author Rebuttal · Authors · 2023-08-09
>
> 1) Page 5's "physics intuition" is hard to understand and unrigorous, and does not seem to do much to alleviate the mentioned "lack of theory" from the Pal et al. approach.
>
> > We appreciate the reviewer's feedback. In our paper, we describe $\Delta r$ in graph theory as the degree of connectivity within a graph, with a higher value signifying increased connectivity due to a smaller non-trivial eigenvalue. This is a commonly held interpretation in graph theory[4]. However, $\lambda$ lacks a universally accepted interpretation. To establish a parallel with $\Delta r$, we interpret $\lambda$ as potential energy, following the approach in [2]. Likewise, similar work such as [3] also implies strong correlation between spectral gaps in estimating the ground-state energy.
> Thus, we interpret a higher $\Delta r$ to  imply a lower overall potential energy, due to smaller non-trivial eigenvalues. As demonstrated in Figure 1 and 2, our goal is to reveal that, even in random architecture searches like our DST, there's a propensity towards architectures that promote expansive models with lower potential energy. This key insight forms the foundation of our paper.
>
> > We highlighted this important limitation of Pal et al. in having insufficient insight and interpretation that acted as a bottleneck to apply Ramanujan bound onto weight adjacency matrix.In hindsight, we recognize that our interpretive efforts may have distracted readers from our main hypothesis. We will clarify these in the final draft.
>
>
> 2) [1] also uses weighted graph regularization to sparsify neural networks weights / Use of weighted graph gap is not so novel.
>
> > We thank  the reviewer for pointing out this work that we will be happy to include in the future revision of this draft. As for the latter point, we have never claimed to be the first  in using weighted graph gap to sparsify neural network weights. For instance, we cited Pal et al. [7], who previously employed weighted graphs  for the similar purpose.
>
> > We thus want to emphasize the contributions of our work: A) the observation that in the context of Ramanujan graph, random evolution of sparse topology is not random; B) the establishment of an almost linear correlation between accuracy and the combined coordinates of the Ramanujan and weighted spectral gap; and C)proposal of new methods, namely PAGS and PEGS,that reliably sample performative masks at initialization, stemming from observations made in points A) and B).
>
> > In specific, **we are the first to link the propensity of sparse neural networks toward Ramanujan expander and weighted spectral gap even when under random growth**. We further empirically demonstrate that maximizing the combined full-spectrum coordinates at initialization can correlate to higher performance.
>
> > We now highlight the key differences between our work and [1]:
> > * [1] uses the weighted spectral gap as a regularizer during training for dense parameters, while we apply our insights at initialization with random weights to identify a performative mask without training. This is an important distinction between Pruning at Training and Pruning at Initialization (PaI).
> > * [1] proposed a conjecture with some proofs relating weighted spectral gap to performance, and utilizing small MLP models in their empirical evidence. We showed in observations of real models that this phenomenon is tangible, and can occur without any specific guidance.
> > * [1] relies on basic MLP layers, and while they argue that their approach can extend to conventional networks, from experience we find this to be technically impractical. Typical linear algorithms, such as those found in Scipy, rely on LAPACK (an efficient and fast linear algorithm package), which utilizes the standard QR algorithm [6] for eigenvalue decomposition. This method involves a series of matrix multiplications, leading to exceedingly high cost while computing  eigenvalues for large dense weight matrices like those in ResNet or VGG. These high costs makes the training time consuming while inflating the memory requirements, thus potentially hindering the applicability of such method. We address these challenges by employing sparse weight matrices proposed by PaI for sparse eigenvalue decomposition, thereby substantially improving the efficiency and scalability.
>
>
> 3) I generally find the paper hard to read. It seems like there is a lot of content and jargon / abbreviations introduced very rapidly.
>
> > Please refer to our global responses.
>
> 4) Experimental improvements are quite modest in Tables 1 and 2.
>
> > Please refer to our global responses.
>
> 5) It would be interesting to more directly optimize for the 4 properties of the coordinate system, instead of simplifying sampling and choosing masks that improve on those properties.
>
> > We appreciate the reviewer's suggestion. However, direct optimization would necessitate full-training, which is beyond the scope of this paper as our interest lies in procuring sparse masks at initialization or with minimal pre-training.
>
>
> 6) Page 3, the use of "tantalizing" is hard to understand for me here. What did you mean by this?
>
> > We use the adjective "tantalizing" to describe our observations, indicating that they are highly interesting and engaging.
>
>
> [1] Tam and Dunson. Fiedler Regularization: Learning Neural Networks with Graph Sparsity. ICML 2020.
>
> [2] https://arxiv.org/abs/1502.04573
>
> [3] https://journals.aps.org/prxquantum/abstract/10.1103/PRXQuantum.3.040327
>
> [4] https://math.uchicago.edu/~may/REU2019/REUPapers/Walchessen.pdf
>
> [6] https://en.wikipedia.org/wiki/QR_algorithm
>
> [7]B. Pal, A. Biswas, S. Kolay, P. Mitra, and B. Basu. A study on the ramanujan graph property of 374 winning lottery tickets. In ICML, 2022.

---

> > ### Comment · Reviewer_i7oA · 2023-08-14
> >
> > We thank the authors for their reply!
> >
> > Thank you for the in-depth comparison to [1], note about the physics intuition, and additional experiments. I do not necessarily mean full backpropagation when I discuss "more directly optimiz[ing]" toward the coordinate system. For instance, one can imagine sampling in a different way such that the expected $\lambda$ of a mask is better.
> >
> > Still, I would like to retain my score, with the caveat that I note that my confidence level is only a 2. I had trouble with the writing of the paper (and also of the comment), but e.g. this may be in part because I am less familiar with this field.

---

> > > ### Author Response · Authors · 2023-08-18
> > >
> > > Dear Reviewer i7oA,
> > >
> > > Thank you for your ongoing feedback. We recognize that there are myriad ways to sample $\lambda$. Nonetheless, our decision to employ a naive sampling approach is purposeful, primarily to underscore our theoretical observations. Importantly, it proves practically efficient in achieving LTH level performance. The heart of our paper lies in our theoretical discovery of the correlation, with new pruning methods presented as proof of concept.
> > >
> > > If any portion of our paper or responses remains unclear, please allow us the chance to offer more insight. If, however, our clarifications have addressed your concerns, we kindly request you to consider adjusting your score. This average score might play a role in the final decision regarding our paper, making your support paramount. We deeply value your time and commitment throughout this review process.
> > >
> > > Warm regards,
> > >
> > > Authors

---

### Author Rebuttal · Authors · 2023-08-09

We thank all our reviewers for their feedback and will try to address each concern individually. While the reviewers tend to have consensus on the novelty and contribution of our work,  there are two common criticisms that we wish to respond globally, namely:
* Insufficient experimentation
* Readability

To address the first concern, we've expanded our experimentation to Tiny-ImageNet. Our results show that PaI, as a baseline, typically underperforms its LTH counterpart. Additionally, we demonstrated efficacy of sampling with PEGS that can significantly enhance performance.

To address the second concern of insufficient readability, primarily due to dense text, excessive jargon, and numerous observations, we will make the following revisions:
* Introduce a table for quick reference to notation definitions and cut-down on number of notations.
* Combine sections 2-4 into "Observations" and "Methodology."
* Clarify our interpretation, describing more simply the relation between performance, topology, and weights.

Within "Observations," we will enhance readability by:
* Streamlining our motivation for using DST to study temporal weight/topology evolution in random fashion.
* Creating a dedicated subsection to clearly describe various metrics and notations, including a summary table, in the context of deep neural networks.
* Adding a section to analyze figures 1-4 and their significance to our central hypothesis.

---

### Decision · Program_Chairs · 2023-09-21

**Decision:**

Accept (poster)

**Comment:**

This paper improves the Ramanujan graph theory towards analyzing deep network pruning-at-initialization (PaI), by viewing a network layer as a weighted graph.

Compared to the prior work (Hoang et. al. 2023), the main improvements made by this paper are: (i) optimizing both the graph’s connection topology and weights (rather than only topology) and proposing the new analytical tool of weighed spectral gap; (ii) turning the weighted graph perspective into a new actionable PaI method, that outperforms previous PaIs and can be on par with LTH at a small fraction of its cost; and (iii) observing how the graph structure evolves in dynamic sparse-to-sparse training, and showing the common routine (pruning by magnitude, then growing randomly) implicitly maximizes Ramanujan graphs. AC also takes an in-detail read of this paper, and personally feels the third point (while very intriguing) had not been dig deep enough in this paper, and only referred to as empirical supporting evidence for the first point. However, the first two points are indeed interesting and well-justified.

After rebuttal, the paper receives four scores of 7, 7, 5, 3. Almost all reviewers agreed that this paper has very rich theoretical contributions as well as analyses; and is written quite logically. Reviewer i70A, who scores the lowest 3, explicitly stated that “…with the caveat that I note that my confidence level is only a 2... this may be in part because I am less familiar with this field.”

The other three reviewers (who scored 7, 7, and 5 after rebuttal) found the common issue of in that the original submission reported only CIFAR-10/CIFAR-100 experiments, hence suggesting a lack of experimental sufficiency. The authors expanded their experimentation to Tiny-ImageNet, during rebuttal. However, Reviewer 62wf still thought ImageNet experiments should have been reported; and in general, all PaI methods underperform other post-training pruning or sparse training methods in practice. The authors also strike back with their arguments.

Reading through the whole discussions, AC is personally familiar with PaI and has the following thoughts. For empirical post-training pruning, upholding experimental validation at ImageNet-level and above is a “must” that AC also agrees on. However, this paper, as well as the generic field of PaI, is more theory-flavor by exploring the “optimal sparse neural architecture" that is independent of any training. Hence, PaI has its unique value, and shall not need outperform more expensive post-pruning pruning methods.

Many PaI papers are published in recent top-tier conferences, such as SNIP (ICLR 2019), SynFLow (NeurIPS 2020), and the unweighted Ramanujan Graph (ICLR'23, which this paper is based on). All those papers meanwhile report experiments on CIFAR-10, CIFAR-100, Tiny-ImageNet; that is because they more prioritize theoretical contributions over extensive experiments. This paper followed the established convention by those PaI papers.

Therefore, by putting it in the context, AC decides that this paper has already done a good job, in delivering what it promised: a solid theoretical foundation for PaI and many meaningful observations. While this paper could be strengthened further by reporting more and larger-scale experiments, the current shape shall suffice to warrant its NeurIPS acceptance.